# *EGFR*-Mutant Non-Small-Cell Lung Cancer at Surgical Stages: What Is the Place for Tyrosine Kinase Inhibitors?

**DOI:** 10.3390/cancers14092257

**Published:** 2022-04-30

**Authors:** Xavier Cansouline, Béatrice Lipan, Damien Sizaret, Anne Tallet, Christophe Vandier, Delphine Carmier, Antoine Legras

**Affiliations:** 1Department of Thoracic Surgery, Tours University Hospital, 37170 Chambray-Lès-Tours, France; xavier.cansouline@etu.univ-tours.fr (X.C.); b.lipan@chu-tours.fr (B.L.); 2Nutrition, Croissance et Cancer, INSERM UMR 1069, University of Tours, 37000 Tours, France; christophe.vandier@univ-tours.fr; 3Department of Pathology, Tours University Hospital, 37170 Chambray-Lès-Tours, France; d.sizaret@chu-tours.fr; 4Platform of Solid Tumor Molecular Genetics, Tours University, 37000 Tours, France; a.tallet@chu-tours.fr; 5Department of Pneumology, Tours University Hospital, 37000 Tours, France; d.carmier@chu-tours.fr

**Keywords:** *EGFR*, NSCLC, adjuvant, neoadjuvant, targeted therapy, resected lung cancer, early stages, tyrosine kinase inhibitors, ADAURA, chemotherapy

## Abstract

**Simple Summary:**

Tyrosine kinase inhibitors are drugs targeting the epidermal growth factor receptor. In lung cancer, they are used to treat advanced *EGFR*-mutant diseases, and more recently, one has been approved for adjuvant therapy. Even though publications on the topic are numerous, conclusions are difficult to interpret and are sometimes contradictory. We therefore reviewed the literature in order to present an overview of up-to-date data regarding the adjuvant and neoadjuvant use of tyrosine kinase inhibitors, with particular attention given to their benefits, proven or expected, as well as what challenges could be faced when entering them as protocols in standard care.

**Abstract:**

The ADAURA trial has been significant for the perception of *EGFR* tyrosine kinase inhibitors (TKIs) as a tool for early stage non-small-cell lung cancer (NSCLC). It produced such great insight that the main TKI, Osimertinib, was rapidly integrated into international guidelines for adjuvant use. However, *EGFR*-mutant NSCLC is a complex entity and has various targeting drugs, and the benefits for patients might not be as clear as they seem. We reviewed trials and meta-analyses considering TKI adjuvant and neoadjuvant use. We also explored the influence of mutation variability and financial evaluations. We found that TKIs often show disease-free survival (DFS) benefits, yet studies have struggled to improve the overall survival (OS); however, the results from the literature might be confusing because of variability in the stages and mutations. The safety profiles and adverse events are acceptable, but costs remain high and accessibility might not be optimal. TKIs are promising drugs that could allow for tailored treatment designs.

## 1. Introduction

In the last decade, the prognosis and treatment of non-small-cell lung cancer (NSCLC) have improved as a result of the discovery of epidermal growth factor receptor (*EGFR*) mutations, which are predictor markers for the response to tyrosine kinase inhibitors (TKIs). Nowadays, at advanced stages of the disease, TKIs are part of the treatment tools, but their place in the early stages remains unclear. Recent studies, such as the ADAURA trial [1], have shown improvements in disease-free survival (DFS) for resected NSCLC in TKI groups, but the impact on the overall survival (OS) remains undefined. Osimertinib is already being introduced into international guidelines. In this review, we aimed to assess the possible place for TKIs in standard care for *EGFR*-mutated NSCLC at the surgical stages in the upcoming years.

## 2. Lung Cancer and *EGFR* Mutations

Lung cancer accounts for 11.4% of all cancer worldwide and was responsible for 1.8 million deaths in 2020 (WHO). Among these, NSCLC is the most common histology [2], and almost one out of three cases are localized upon diagnosis. For these cases, the gold-standard treatment is surgical resection [3]—anatomical resection through lobectomy or sub-lobectomy, or segmentectomy in the very early stages [4], combined with mediastinal lymphadenectomy [5]. Surgery is associated with peri-operative chemotherapy in cases of nodal involvement or with a tumor size of >4 cm [6]. Chemotherapy slightly improves OS (absolute benefits of 3.9% and 5.4% after 3 and 5 years, respectively) and DFS (absolute benefits of 5.8% and 5.8% after 3 and 5 years, respectively) outcomes, but the recurrence rate remains high—up to 50% after 5 years [7]. At an advanced stage, lung cancer is associated with a poor prognosis, and standard care is represented by platinum-based chemotherapy, with small improvements in OS and progression-free survival (PFS) [8] and high toxicity profiles [9].

The discovery of oncogenic driver mutations allowed for the development of targeted therapies, which rapidly became the standard treatment for eligible patients as a result of their major improvement outcomes [10,11]. These mutations are numerous; however, *EGFR* mutations are the most represented, and are present in 15% of lung adenocarcinomas in the USA and up to 60% in Asian females [2].

The epidermal growth factor receptor is a transmembrane receptor from the tyrosine kinases family HER/erbB; it is involved in proliferation, angiogenesis, and apoptosis inhibition. Its gene is located on the short arm of chromosome 7 (7p11.2), and 93% of mutations of interest are found within exons 18–21, which code for the tyrosine kinase domain responsible for abnormal activation in cancerous cells [12]. Of these mutations, 90% are represented by exon 19 deletions (Ex19del) and exon 21 L858R mutations [13], which have been proven to be sensitive to TKI drugs [14]. On the contrary, the exon 20 T790M mutation produces resistance to targeted therapy, and is mainly an acquired mutation after clonal selection as a result of drug administration [11]. While exon 20 T790M confers resistance to first- and second-generation *EGFR* TKIs, it can be overcome by third-generation TKIs (i.e., Osimertinb) [15]. Another frequent (around 5% of all *EGFR* mutations) [16] TKI-resistant mutation family is found in exon 20 insertions. Although these mutations currently confer de novo resistance for most available TKIs [17], Osimertinib may provide a moderate response [18], and Mobocertinib, a new drug, is in the early stages of evaluation for this particular indication [19]. Less common mutations are also found on exons 18, 20, and 21, and a small amount of patients have been found to have complex mutations, associated with two or more mutations [20]. *EGFR*-mutant (*EGFR*m) NSCLC mainly occurs in nonsmoking Asian females with adenocarcinoma [21].

The development of three generations of *EGFR* inhibitors has shown increased outcomes [22] and acceptable security profiles, providing them a place as a first-line treatment for advanced *EFGR*m NSCLC [23]. These improvements have provided insight so as to enhance the prognosis of patients with localized stages in adjuvant and neoadjuvant therapy.

## 3. TKIs as Neoadjuvant Treatment

TKIs might be able to be used as neoadjuvant treatment; however, the topic has been poorly explored. Some phase II trials have found potential improvements with Erlotinib; the main results from these trials are summarized in Table 1.

Chen et al. [24] compared Erlotinib versus Pemetrexed−Cisplatin chemotherapy in a randomized trial with 86 patients and found an improved objective response rate (ORR)/risk ratio (RR) of 1.53 (95% CI: 1.03–2.27) and operation rate (OpR)/RR of 1.08 (95% CI: 0.9–1.27). Zhong et al. [26] also compared Erlotinib to chemotherapy in a 24-patient non-randomized trial; their results contradicted other studies because the OS was in favor of chemotherapy, with a hazard ratio (HR) = 1.79 (95% CI: 0.73–4.39).

Xiong et al. [27] performed a trial with 31 patients, and compared Erlotinib treatment for *EGFR*m patients versus chemotherapy treatment for *EGFR*-wild-type (*EGFR*w) patients. Their findings found an improved ORR (67% vs. 19%), OpR (12/15 vs. 8/16), pathological response rate (67% vs. 38%), and OS (51 vs. 20.9 months) in the Erlotinib group. They also found an improved tumor diameter reduction (35% vs. 16%) with Erlotinib.

Recently, Zhong et al. [25] published EMERGING CTONG, a controlled trial that included 72 patients receiving Erlotinib or chemotherapy; it showed an encouraging ORR (primary outcome) of 54.1% (95% CI: 37.2–70.9%) versus 34.3% (95% CI: 17.7–50.8%), odds ratio (OR) of 2.26 (95% CI: 0.87–5.84; *p* = 0.092), and node down staging (secondary outcome, 10.8% vs. 2.9%, *p* = 0.185), without reaching statistical significance. The PFS (secondary outcome) was significantly better: 21.5 months (95% CI: 16.7–26.3 months) versus 11.4 months (95% CI: 7.3–15.5 months), and HR of 0.39 (95% CI: 0.23–0.67; *p* = 0.001). The OS (secondary outcome) was comparable in both groups: 45.8 versus 39.2 months (HR = 0.77, 95% CI: 0.41–1.45; *p* = 0.417). These results might have been attenuated by the sample size, slow rate of accrual, and short time of exposure. The safety profile was good with no grade 3/4 toxicities in the Erlotinib group. The most common adverse events (AEs) were rash, diarrhea, and cough, as described before. These results warrant more investigations with larger cohorts in order to give stronger evidence of the improvement provided by TKI in neoadjuvant therapy.

In 2021, Chen et al. [29] published a meta-analysis collecting data from the five trials (three RCT [25,28,29] and two non-RCT [26,27]) available comparing Erlotinib to chemotherapy as neoadjuvant treatments for *EGFR*m stage IIIA NSCLC patients; they found a trend in favor of Erlotinib in OS (HR = 0.74, 95% CI: 0.43–1.27) and PFS (HR = 081, 95% CI: 0.27–2.44). The ORR (RR = 1.70, 95% CI: 1.35–2.15), progression rate (RR = 0.64, 95% CI: 0.34–1.19), and OpR (RR = 1.13, 95% CI: 1.01–1.26) were in favor of TKIs in all of the studies included, resulting in significant improvements when using Erlotinib. The effect on OpR was lost when excluding the data from non-RCT trials. The toxicity was greater in the Erlotinib group (RR = 0.50, 95% CI: 0.34–0.75), mainly presenting as skin rash, but not significantly different when excluding the non-RCT data (RR = 0.53, 95% CI: 0.26–1.08).

Gefitinib has not been evaluated as much. Rivzi et al. [30] enrolled 50 patients with stages I–II NSCLC in a single-arm trial; patients from a population enriched for *EGFR* mutations received TKI for 21 days before surgery. The authors reported a radiographic response (>25% reduction in bidimensional measurement) for 21 patients (42%), in which 17 were *EGFR*m. Nine patients were upstaged and seven were downstaged, but it was not detailed from which group they were. DFS did not significantly differ when stratified according to *EGFR* status or adjuvant TKI usage. The OS data were incomplete and, to our knowledge, were never reported. No grade IV AEs were reported and only one grade III rash occurred during the pre-operative period.

Zhang et al. [31] published a trial of 35 patients with stages II−IIIA *EGFR*m NSCLC who received Gefitinib 42 days before surgery. They found an ORR of 54.5% (95% CI: 37.7–70.7) and a rate of major pathologic response (MPR) of 24.2% (95% CI: 11.9–40.4) using the RECIST criteria. The median DFS was 33.5 months (95% CI: 19.7–47.3) and the median OS was not achieved. They also found that patients with MPR had a significantly longer median DFS (68 vs. 25.3 months, *p* = 0.019) and a trend for a longer OS (*p* = 0.134).

NEOADAURA [32], a phase III trial, is currently ongoing, comparing Osimertinib with or without chemotherapy versus chemotherapy alone in resectable stages II–IIIB N2 *EGFR*m (Ex19del or L858R) NSCLC. The trial is ongoing and might not end before 2024 (estimated final data collection date for primary outcome measure). Of note, nine single-arm phase II trials are currently ongoing to explore new schemes [33,34,35,36,37,38,39,40,41]. These studies are summarized in Table 2.

## 4. TKI as Adjuvant Treatment

TKI was first described as an adjuvant therapy by Tsuboi et al. [42] in 2005 in a 38-patient phase III trial comparing Gefitinib to a placebo. They provided safety and feasibility data, but no survival data because of early trial termination due to a controversy involving a rising rate of interstitial lung disease (ILD) in patients treated with Gefitinib [43] (see safety profile below).

Since this publication, a dozen clinical comparisons [44,45] have been conducted, with four phase II [46,47,48,49] and seven phase III [1,42,50,51,52,53,54] randomized trials. When using TKI, most of these trials showed improved DFS compared with a placebo, with an HR ranging from 0.2 to 0.61, with statistical significance for all of the trials (Table 3), except for Feng et al. [48], where the statistical power may have been limited by the sample size (39 patients) and follow-up period (24 months, median DFS not reached).

In 2015, Kelly et al. [50] published the RADIANT trial, a randomized, double-blind, placebo-controlled phase III international and multi-center trial that included 973 patients. They compared Erlotinib with a placebo in patients with *EGFR*-expressing tumors, but not necessarily *EGFR*-mutated tumors (n = 161). For the *EGFR*m patients, DFS (as a secondary endpoint) was better (HR = 0.61, 95% CI: 0.38–0.98) with Erlotinib, but the primary outcome was not reached (DFS in the ITT population, HR = 0.90; 95% CI: 0.74–1.10; *p* = 0.324).

It is not clear yet if Gefitinib should be used as an adjuvant therapy. The findings of Goss et al. [52] (BR19 trial, Phase III, 2013) on DFS were not in favor of Gefitinib (HR = 1.84, 95% CI: 0.44–7.73), but the study was stopped early because of a lack of supportive evidence regarding the efficacy of this drug in trials for advanced stages [55]. Zhong et al., in CTONG 1104/ADJUVANT [53] (2018), compared Gefitinib versus Vinorelbine plus Cisplatin (VP), and showed an improved median DFS in the TKI group of 30.8 months (95% CI: 26.7–36.6) versus 19.8 months (95% CI: 15.4–23.0), with an HR of 0.51 (95% CI: 0.36–0.72, *p*< 0.001). The long-term data did not show any benefit, with 3-year DFS rates of 39.6% (TKI) and 32.5% (VP) (*p* = 0.316) and 5-year DFS rates of 22.6% (TKI) and 23.2% (VP) (*p* = 0.928). The Kaplan−Meier curves crossed at 60 months, but the risk at this point represented only 10% of the original contingent, exposing this result to a power insufficiency [56]. Tada et al. [57] recently published IMPACT WJOG6410L, a phase III randomized trial with 234 patients with stages II–III, *EGFR*m NSCLC. The patients either received 2 years of adjuvant Gefitinib or four cycles of Cisplatin−Vinorelbine adjuvant chemotherapy. Neither DFS nor OS were different between the two arms (DFS HR = 0.92; 95% CI: 0.67–1.28; *p* = 0.63) (OS HR = 1.03; 95% CI: 0.65–1.65; *p* = 0.89).

In 2021, He et al. [54] published the EVIDENCE trial—a phase III randomized trial comparing Icotinib (highly selective first-generation *EGFR* TKI) to chemotherapy in 322 patients. Their findings were consistent with previous trials, with an improved DFS of 47 months (95% CI: 36.4–not reached) with Icotinib versus 22.1 months (95% CI: 16.8–30.4) with chemotherapy (HR = 0.36; 95% CI: 0.24–0.55; *p* < 0.0001). The 3-year DFS was also better (63.9% (95% CI: 51.8–73.7) vs. 32.5% (95% CI: 21.3–44.2)), but the OS data were incomplete and not published. To date, it is the largest trial on Icotinib in an adjuvant setting.

The ADAURA [1] study, a phase III trial, compared 3-year adjuvant Osimertinib versus a placebo in stages IB−IIIA (seventh TNM classification) *EGFR*m (Ex19del or L858R) NSCLC. It is currently the largest cohort to date, with 682 patients enrolled. Adjuvant chemotherapy before targeted therapy was allowed, but not mandatory. The primary endpoint, DFS at 24 months for stages II−IIIA, was shown to be dramatically better for TKI (HR = 0.17; 99.06% CI: 0.11–0.26), with an 80% reduction of risk of recurrence after 2 years. In the overall population (stages IB−IIIA, with stage IB accounting for 32% of the cohort), the same effect was shown (HR = 0.20; 99.12% CI: 0.14–0.30; *p*< 0.001). In the subgroup analysis, the effect was consistent and progressive according to disease stage, with an HR of 0.39 (95% CI: 0.18–0.76) in stage IB, HR of 0.17 (95% CI: 0.08–0.31) in stage II, and HR of 0.12 (95% CI: 0.07–0.2) in stage IIIA. The use of precedent adjuvant chemotherapy also slightly enhanced DFS (HR = 0.16 with chemo vs. 0.23 without, 95% CI: 0.10–0.26 and 0.13–0.40, respectively). They also reported an 82% reduction in the risk of death or central nervous system (CNS) recurrence (HR = 0.18; 95% CI: 0.10–0.33), but the median CNS DFS was not achieved, and this result should be taken with caution. According to these results, the independent data monitoring committee decided to unblind the trial early. At publication, the OS data were incomplete, and the follow-up is still ongoing as well as the quality-of-life evaluation. Although these results are impressive, this study might be biased in its design. Firstly, a PET−CT scan was not mandatory at baseline staging. Secondly, brain imaging was not updated before TKI introduction, even though the delay between surgery and randomization was allowed to be up to 26 weeks when adjuvant chemotherapy was administered (10 weeks without chemotherapy). Furthermore, there are no data for assessing the quality of surgical resection (lymphadenectomy extend and lymph node capsular effraction, complete or incomplete resection, operative approach). The duration of treatment was 3 years, which is longer than in other trials; however, it is not clear if longer therapy would improve the outcomes [58]. These limitations might have resulted in an underestimation of the staging at randomization, lowering the control of this possible confounding factor. Nevertheless, ADAURA largely contributed to the inclusion of Osimertinib in North American guidelines as an adjuvant treatment for NSCLC with *EGFR* Ex19del or L858R mutations [59], and in Europe for stages IB−IIIA with the same mutations [60,61]. Of note, ADAURA used the seventh TNM classification with stage IB with tumors of at least 4 cm, which are now classed stage II in the eighth TNM [62]. However, these guidelines would be applied to new stage IB tumors, even though this patient group was not evaluated.

Concerning OS, only the EVAN [49] phase II trial was able to prove a statistical benefit of Erlotinib over chemotherapy as a secondary endpoint (HR = 0.165, 95% CI: 0.047–0.579), but this result might be immature because the median OS was not reached for both groups at the data cutoff and there were fewer death events compared with the study discontinuation events. In other trials, the results were heterogeneous from one study to another (HR ranging from 0.37 to 3.16, *p* > 0.05).

Soon, the French Intergroup of Thoracic Cancerology (IFCT) will conduct the ROSIE trial, which aims to identify clinico-pathological and molecular descriptions associated with the probability of relapse after Osimertinib adjuvant exposure in completely resected stages pIIA–IIIA (TNM staging 8th) *EGFR*m NSCLC, and to describe the clinical, pathological, and molecular characteristics upon relapse during or after Osimertinib treatment. The protocol will include plasmatic circulating tumor DNA (ctDNA) level monitoring, as well as tumor genomic sequencing at baseline and at relapse.

Regarding future publications, we considered the ALCHEMIST [63] trial, a nationwide study evaluating Erlotinib versus a placebo in stages IB (>4 cm)−IIIA *EGFR*m NSCLC. The primary outcome is OS and investigators plan to include 450 patients, with a complete follow-up of 6 years after ending treatment. For more details regarding the ongoing comparative trials, see Table 4.

Ten meta-analyses [45,67,68,69,70,71,72,73,74,75] assessed that DFS was better with TKI than without [69] (HR ranging from 0.38 to 0.86 with statistical significance in all ten publications). However, only two of five found an improvement in OS—Tang et al. [73] in 2019 with an OR of 0.63 (95% CI: 0.46–0.86) and Yin et al. [75] in 2021 with an HR of 0.62 (95% CI: 0.45–0.86), but they included non-randomized trials [76,77,78]. One of the latter trials highlighted the heterogeneity of the stage and mutation repartition from one trial to another [67], which could be confounding factors. We found that the patient pool from stages IB to IIIA regrouped a very wide variety of cancers with an inconstant prognosis, and that mutation variability could also interfere with the drug response (see below—Impact of *EGFR* Mutation). Thus, entering Osimertinib into routine protocol will surely provide data, but its administration should be monitored to define who should receive it and who should not. Of note, unsurprisingly, TKI showed no benefits when used on *EGFR*-wild-type NSCLC.

## 5. Impact of *EGFR* Mutation and Co-Mutations

Although Ex19del and L858R are the most frequent mutations and markers for TKI sensitivity, they might not be equal. In advanced cancers, Lee et al. [79] revealed in a meta-analysis that the PFS for tumors with the Ex19del mutation could be up to 50% greater (HR = 0.24, 95% CI: 0.20–0.29) than for tumors with L858R (HR = 0.48, 95% CI: 0.39–0.58). In adjuvant therapy, the subgroups analysis data from ADAURA, EVAN, ADJUVANT, and EVIDENCE were consistent with these results, with a trend towards a better DFS in patients with Ex19del. There are also dozens of rare mutations that could produce the opposite reactions from one or another drug [80], highlighting the challenge of precise genomic sequencing in order to allow for personalized treatment plans. Furthermore, as TKI could promote the emergence of a resistant clone under selection pressure, tumor re-sequencing upon recurrence might be an important step in adaptive therapy.

Co-mutations may also be predictive for drug response. Liu et al. [81] reviewed the genomic profiles from the ADJUVANT trial and found five predictive biomarkers (TP53 exon4/5 mutations; RB1 alterations; and copy number gains of MYC, NKX2-1, and CDK4). They were able to define three subtypes of patients with very different responses to Gefitinib in each group—sometimes opposite. These data show the complexity of genomic impact and the uncertainty of the drug response from one patient to another. In the future, as TKIs are becoming part of standard treatment, very careful routine patient data collection will be needed to further document this variability and to design better tailored treatment paths.

## 6. Safety

As TKIs struggle to show an impact on OS, safety, AEs, and quality of life (QoL) appear to be key points for patients’ and clinicians’ acceptability. At the beginning of its use, Gefitinib was shown to cause interstitial lung disease, which could severely injure the lung capacity [43]. Investigations led to the conclusion that this AE, which can occur for any *EGFR* TKI, is rare and is more likely to occur in Japanese populations [82]. ILD is usually reversible after drug discontinuation, and steroids and reliable drugs can replace TKIs [83]. Dacomitinib and Afatinib have a higher toxicity than Icotinib, Osimertinib, and Gefitinib, but Osimertinib seems to provide the highest risk for ILD [84]. Most trials and meta-analyses have provided safety data—TKIs are more likely to cause AEs than a placebo, but much less than adjuvant chemotherapy [68]. Grade III−IV AEs are extremely rare; the most common AEs are diarrhea, paronychia, mucocutaneous symptoms (skin rash, dry skin, pruritus, mouth ulceration, etc.), and cough. Treatment discontinuation and dose reductions due to AEs are less than in the chemotherapy groups in the ADJUVANT, EVIDENCE, and EVAN trials. ADAURA’s QoL was reported and showed no difference between Osimertinib and a placebo [85]. Finally, as TKIs are oral drugs with a single daily dose, adhesion and acceptation will be better than for intravenous treatments.

## 7. Cost Effectiveness and Accessibility

With every new drug class being routinely used, the financial aspect for health organizations is considered. Lemmon et al. [86] performed a cost-effectiveness model using the ADAURA data and calculated an incremental cost-effectiveness ratio (ICER) for Osimertinib of about USD 317,000 per quality-adjusted life year (QALY) gained (E.D: “One quality-adjusted life year (QALY) is equal to 1 year of life in perfect health”, NICE Glossary). Choi et al. [87] did the same by detailing cost according to stage. They found a better incremental benefit in the stage II group than in the IB and IIIA groups, but with higher a incremental cost−utility ratio (ICUR) than Lemmon, with around USD 1.2 million (Groups IB–IIIA) and USD 636,000, per life year gained. These models are limited by the incompleteness of the OS data, and their outcomes (QALY gain vs. life year gain) may not be comparable. The implementation of adjuvant therapy in standard care will have substantial costs that will warrant more accurate cost-effectiveness evaluations. These costs, associated with the lack of evidence of the benefit on OS, might limit their use to wealthier countries and patients.

## 8. Discussion

TKIs are promising drugs, yet there are still some crucial points to explore. First is the question of an optimal treatment duration [54]. It has been highlighted that most recurrence in TKI arms occurs after treatment discontinuation, with DFS perhaps being dependent on the exposition time. Then, it could be hypothesized that TKIs will not cure the disease, but only keep cancerous cells in a dormant state, while slowly selecting resistant clones, resulting in a delayed but inexorable recurrence [88,89]. This could explain why the benefits of OS are so tricky to find. To date, the duration of treatment is arbitrary and relies on evidence from shorter DFS with shorter treatment times (i.e., ADJUVANT vs. ADAURA); however, more investigations are warranted to develop and define this concept. Indeed, the merits of adjuvant TKIs may also be considered for very early stages—in ADAURA, the stage IB DFS at 24 months was 88% and 71% with the TKIs and a placebo, respectively. Therefore, we do not know how the recurrence was rechallenged, especially in the TKI group, where the clonal selection could make it untreatable with other TKIs. Ultimately, we cannot know if these patients really benefited from Osimertinib. One might wonder if active monitoring and TKI challenging at relapse would not be a more efficient strategy, with a delayed use of chemotherapy (and thus preserved QoL) and wiser money spending.

Second, while TKIs are introduced as an adjuvant treatment more and more frequently by national and international guidelines, evidence of their survival improvements needs to be more underpinned. With a wide range of mutations, stages, and prognoses, it is necessary to more accurately explore which groups could benefit more from TKI adjuvant therapy. Their use needs to be managed by genomic detection of appropriate mutations and co-mutations. This detection, associated with a high cost of treatment, may slow down their inclusion into standard care and accessibility for all. With good safety profiles and few AEs, TKIs could enter adjuvant therapy panels with a good patient adhesion expected. Even without evidence of the benefit of survival over chemotherapy, this point may be important in patients for whom an adjuvant treatment is formally indicated, but who are unfit to receive chemotherapy. In these cases, TKIs could be an alternative to chemotherapy, with better outcomes than abstention, while maintaining their QoL.

Eventually, these drugs might also have a place as neoadjuvant treatments, but this statement needs to be supported by more accurate data and well-conducted trials.

We aimed to highlight the challenges regarding what could be the future of oncologic practice: individualized therapy design, with clinical, pathological, and genomic profiles very precisely defined at baseline and at relapse. There is still a lot to understand about *EGFR*m NSCLC, and as the amount of data needed will be huge, we can only place our hopes in the usefulness and completeness of national and international databases to help investigators find ways to relentlessly fight this disease [90,91]. We believe that the success of this new era will be dependent on the meticulous collection and sharing of patient data.

## 9. Conclusions

*EGFR* TKIs in adjuvant therapy clearly improve DFS, but not, to date, OS. The optimal duration of adjuvant treatment and its real place among other therapeutic tools should be defined. In neoadjuvant therapy, *EGFR* TKIs may improve ORR and operation rates. Their safety and toxicity are good, but the high cost of treatment may restrain accessibility. The understanding the effects of the numerous mutations must be improved for individualized drug assignment and tailored treatment plans.

## Figures and Tables

**Table 1 cancers-14-02257-t001:** Results of trials considering neoadjuvant therapy.

Name, Author Year	Design	Patients	Stage	HR (*p*)					N+ Down-Staging (*p*)
				ORR	OpR	PRR	PFS	OS	
Chen 2018 [24]	**Phase II** **Erlotinib vs. chemo (P + C)**	*EGFR*m (86)	IIIA	1.53 (<0.05)	1.08 (>0.05)	1.55 (<0.05)	NA	NA	NA
CTONG 1103, Zhong 2019 [25]	**Phase II** **Erlotinib vs. chemo (G + C)**	*EGFR*m (72)	IIIA N2	2.26 (0.092)	NA, 83.8% vs. 68.6% (0.129) ^†^	NA, 9.7% vs. 0% ^†^	0.39 (<0.001)	0.77 (0.417)	10.8% vs. 2.9% (0.185) ^†^
Zhong 2015 [26]	**Phase II** **Erlotinib vs. chemo (G + C) ***	*EGFR*m (12) *EGFR*w (12)	IIIA N2	NA, 58.3% vs. 25% (0.18) ^†^	NA, 6/12 vs. 7/12 ^†^	NA	2.26 (0.071)	1.79 (0.201)	25% vs. 25% (1) ^†^
Xiong 2019 [27]	**Phase II** **Erlotinib vs. chemo ***	*EGFR*m (15) *EGFR*w (16)	IIIA	NA, 67% vs. 19% ^†^	NA, 12/15 vs. 8/16 ^†^	NA, 67% vs. 38% ^†^	NA, 12.1 vs. 11 ^†,‡^	NA, 51 vs. 20.9 (0.12) ^†,‡^	NA
Ning 2019 [28]	**Erlotinib vs. chemo (P + C) ***	*EGFR*m (53) *EGFR*w (53)	IIIA	NA, 35/53 vs. 22/53 ^†^	NA, 46/53 vs. 43/53 ^†^	NA	NA	NA	NA

* Chemotherapy in the *EGFR*w group, Erlotinib in the *EGFR*m group; ^†^ data from the tyrosine kinase inhibitors groups vs. data from the chemotherapy groups; ^‡^ Values in months; HR (*p*): ORR, OpR, PRR, PFS, and OS are expressed as hazard ratio, *p*: *p*-value; ORR: objective response rate; OpR: operation rate; PRR: pathological response rate; PFS: progression-free survival; OS: overall survival; Chemo: chemotherapy; P: Pemetrexed; C: Cisplatin; G: Gemcitabine; *EGFR*m: *EGFR* mutant; *EGFR*w: *EGFR*-wild-type; NA: not available.

**Table 2 cancers-14-02257-t002:** Ongoing phase II trials considering neoadjuvant therapy. All these data are available on https://www.clinicaltrials.gov/ (accessed on 30 March 2022).

Name, Trial Number	Tested Drug	Treatment Plan	Adjuvant TKI	Primary Outcome	Stage	*n*
NCT04685070 [33]	Almonertinib	8–16 weeks	up to 40 weeks	ORR	Stage III	56
NCT04201756 [34]	Afatinib	8–16 weeks	1 year	ORR	Stage III	47
NCT03749213 [35]	Icotinib	8 weeks	2 years	ORR	IIIA N2	36
NCT03349203 [36]	Icotinib	8 weeks	2 years	ORR	IIIB or oligometastasis	60
Neolpower, NCT05104788 [37]	Icotinib + chemo	12 weeks	NA	MPR	IIA−IIIB	27
NCT02820116 [38]	Icotinib	8 weeks	2 years	Complete resection rate	IIIA−IIIB	67
NOCE01, NCT05011487 [39]	Osimertinib + chemo	60 days	NA	Complete lymph node clearance rate *	IIIA−IIIB N2	30
NCT03433469 [40]	Osimertinib	4–10 weeks	No	MPR	I−IIIA	27
ASCENT, NCT01553942 [41]	Afatinib + radio + chemo	8 weeks	2 years	Response rate	IIIA	30

* The ratio of ypN0 percentage after resection. Chemo: chemotherapy; Radio: radiotherapy; ORR: objective response rate; MPR: major pathologic response; DFS: disease-free survival; OS: overall survival; *n*: expected patient number; NA: not available.

**Table 3 cancers-14-02257-t003:** Results of the main trials considering adjuvant therapy.

Name, Author Year	Design	*EGFR* Status (*n*)	Stages	HR DFS (*p*)	HR OS (*p*)
Tsuboi 2005 [42]	Phase III Gefitinib vs. placebo	UP (38)	IB−IIIA	NA	NA
BR19, Goss 2013 [52]	Phase III Gefitinib vs. placebo	UP (503) *EGFR*m (15)	IB−IIIA	1.84 (0.395)	3.16 (0.15)
Li 2014 [46]	Phase II Chemo (P + C) + Gefitinib vs. chemo alone	*EGFR*m (60)	IIIA (N2) (5th TNM)	0.37 (0.014)	0.37 (0.076)
Feng 2015 [48]	Phase II chemo + Icotinib vs. chemo alone	*EGFR*m (39)	IB−IIIA	NA, 21 vs. 16 (0.122)	NA
RADIANT, Kelly 2015 [50]	Phase III Erlotinib vs. placebo	*EGFR*exp (973) *EGFR*m (161)	IB−IIA (6th TNM)	0.61	1.09 (0.815)
EVAN, Yue 2018 [49]	Phase II Erlotinib vs. chemo (V + P)	*EFGR*m (102)	IIIA	0.268 (<0.0001)	0.165 (0.0013)
ADJUVANT, Zhong 2018 [53]	Phase III Gefitinib vs. chemo (V + P)	*EGFR*m (222)	II−IIIA (N1-2)	0.51 (0.001)	0.92 (0.674)
ADAURA, Wu 2020 [1]	Phase III Osimertinib vs. placebo	*EGFR*m (682)	IB−IIIA (7th TNM)	0.2 (<0.001)	NA
IMPACT, Tada 2021 [51]	Phase III Gefitinib vs. chemo (V + P)	*EGFR*m (234)	II−IIIA	0.92 (0.63)	1.03 (0.89)
EVIDENCE, He 2021 [54]	Phase III Icotinib vs. chemo (V + P)	*EGFR*m (322)	II−IIIA (7th TNM)	0.36 (<0.0001)	0.75 (>0.05)
* SELECT, Pennel 2019 [47]	Phase II Erltonib after chemo (comparison to historical data)	*EGFR*m (100)	IA−IIIA (7th TNM)	NA, 88% vs. 76% (0.047) ^†^	NA

* All of the presented studies are RCT, except for SELECT, which is a single-arm trial. ^†^ Data in tyrosine kinase inhibitors groups vs. data in chemotherapy groups. Chemo: chemotherapy; V + P: Vinorelbine plus Cisplatin; P + C: Pemetrexed plus Carboplatin; UP: unselected patients; *EGFR*exp: *EGFR* expressing; *EGFR*m: *EGFR* mutant; NA: not available.

**Table 4 cancers-14-02257-t004:** List of ongoing comparative trials. All these data are available on https://www.clinicaltrials.gov/ (accessed on 30 March 2022).

Name, Trial Number	Design	Setting	Treatment Arms	Primary Outcome	Stages	*n*
NeoADAURA, NCT04351555 [32]	Phase III, randomized, controlled, multi-center, 3-arm trial	Neo- adjuvant	Osimertinib + chemo vs. placebo + chemo vs. Osimertinib	MPR	II−IIIB N2	328
ADAURA2, NCT05120349 [64]	Phase III, randomized, controlled, multi-center, international, 2-arm trial	Adjuvant	Osimertinib vs. placebo	DFS	IA2, IA3	380
ICTAN, NCT01996098 [65]	Phase III, randomized, open label, multi-center, 3-arm trial	Adjuvant, after chemo	Icotinib 6 months vs. Icotinib 12 months vs. no treatment	DFS	IIA −IIIA	318
APEX, NCT04762459 [66]	Phase III, randomized, open label, multi-center, 3-arm trial	Adjuvant	Almonertinib vs. Almonertinib + chemo vs. chemo	DFS	II −IIIA	606
ALCHEMIST, NCT02193282 [63]	Phase III, randomized, controlled, nationwide, multi-center, 4-arm trial	Adjuvant	Erlotinib (blinded) vs. placebo (blinded) vs. Erlotinib (unblinded) vs. placebo (unblinded)	OS	IB (≥4 cm) −IIIA	450

Chemo: chemotherapy; MPR: major pathologic response; DFS: disease-free survival; OS: overall survival; *n*: expected patient number.

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
