# Peer review of "EGFR-Mutant Non-Small-Cell Lung Cancer at Surgical Stages: What Is the Place for Tyrosine Kinase Inhibitors?"

_cancers, 2022, doi:10.3390/cancers14092257_

Round 1
Reviewer 1 Report
Cansouline and collaborators aimed to review the available data about the role of EGFR TKIs for the management of patients with resectable EGFR-mutated NSCLC.
This is a timely topic of clinical and academic interest, anyway I would advise some relevant revisions.
- The manuscript needs an extensive editing of English language and style in the paragraphs 1, 2, and 3, before to be considered for publication.
- Furthermore, in the same sections, the sentences in the following lines should be corrected/implemented:
- 49-50: Nodal involvement is not the only criteria considered for neo-/adjuvant chemotherapy in resectable NSCLC.
- 67-69: T790M mutation in ex20 confers resistance to first- and second-generation EGFR TKIs, while it can be overcome by third-generation TKIs (i.e. osimertinb).
- 129: I would specify that TKIs improved DFS compared to placebo
- Table 1 is really difficult to understand. It should be edited in order to improve readability. The role of “HR (p)” above the table is not clear. Author, Design, Num of patients and Stages sections should be moved to first columns on the left. Ref. 22 in table 1 is the same as Ref. 24 used in the text.
- Among the studies considered in the neoadjuvant setting, there is no mention about those evaluating gefitinib. I would consider Zhang et al. 2020 doi: 10.1016/j.jtcvs.2020.02.131 and Rizvi et al. 2011 doi: 10.1158/1078-0432.CCR-10-2102
- The paragraphs 4-9 are better written. Minor spell check is required.
- Finally, I would advise (however, I leave the final decision with the authors):
- to expand the discussion: it appears too general, while some points could be addressed more carefully (i.e. duration of adjuvant TKI, role of adjuvant TKI in patients which are unfit for chemotherapies…)
- a table summarizing ongoing clinical trials in this setting could add value to the whole review
- In the title, I would use the more common form “Non-Small Cell Lung Cancers” instead of “Non-Small Cells Lung Cancers”
Author Response
Dear reviewer,
Thank you for you review and comments. Due to your reply, we made (among other suggested revisions) the following modification:
- The manuscript needs an extensive editing of English language and style in the paragraphs 1, 2, and 3, before to be considered for publication: These paragraphs were completely revised in their form.
- Furthermore, in the same sections, the sentences in the following lines should be corrected/implemented:
- 49-50: Nodal involvement is not the only criteria considered for neo-/adjuvant chemotherapy in resectable NSCLC: Precisions added.
- 67-69: T790M mutation in ex20 confers resistance to first- and second-generation EGFR TKIs, while it can be overcome by third-generation TKIs (i.e. osimertinb): This precision was added, along with references and the paragraph was slightly developed.
- 129: I would specify that TKIs improved DFS compared to placebo: Precision added.
- Table 1 is really difficult to understand. It should be edited in order to improve readability. The role of “HR (p)” above the table is not clear. Author, Design, Num of patients and Stages sections should be moved to first columns on the left. Ref. 22 in table 1 is the same as Ref. 24 used in the text: Table 1 is now reorganized. We added a precision on the meaning of “HR (p)” in the footer. We resynchronized the bibliography.
- Among the studies considered in the neoadjuvant setting, there is no mention about those evaluating gefitinib. I would consider Zhang et al. 2020 doi: 10.1016/j.jtcvs.2020.02.131 and Rizvi et al. 2011 doi: 10.1158/1078-0432.CCR-10-2102: We added a paragraph about these studies and their results.
- The paragraphs 4-9 are better written. Minor spell check is required: We corrected orthograph and syntax.
- Finally, I would advise (however, I leave the final decision with the authors):
- to expand the discussion: it appears too general, while some points could be addressed more carefully (i.e. duration of adjuvant TKI, role of adjuvant TKI in patients which are unfit for chemotherapies…): We expanded discussion and insisted on duration of treatment and its indications.
- a table summarizing ongoing clinical trials in this setting could add value to the whole review: We added 2 tables about ongoing trials.
- In the title, I would use the more common form “Non-Small Cell Lung Cancers” instead of “Non-Small Cells Lung Cancers”: Title corrected.
You will find a revised version of the manuscript that we consider to be improved. We humbly hope this new version will fill your expectations and that this review will arouse the reader’s interest.
Regards.
Xavier Cansouline.

Reviewer 2 Report
In the present manuscript, the authors successfully review and discuss the use of Tyrosine Kinase Inhibitors as neoadjuvant and adjuvant treatment based on the data of the main clinical trials of the field. They provide information regarding the stages of the disease, the EGFR status, the specific inhibitor used in each study and finally the outcome based on the differences on the Disease free survival (DFS) and overall survival (OS). In the next section, the authors discuss another important aspect, the specific EGFR mutations and the co-mutations that may exist and could be predictive for the drug response. This comment highlights the importance and the necessity of personalized treatment plans in these patients.
Overall, this is a very interesting review and I support the publication in its present form.
Minor issue:
The authors should improve the format of Table 1.
Author Response
Dear reviewer,
Thank you for you review and comments. Due to your reply, we made (among other suggested revisions) the following modification:
- The authors should improve the format of Table 1: Table 1 has been reorganized.
We are honored to see that our manuscript filled your expectations, and we humbly hope this review will arouse the reader’s interest. You will find a revised version of the manuscript that we consider to be improved.
Regards.

Reviewer 3 Report
Authors present a well written and exhaustive review of medical literature on the use of tyrosine kinase inhibitors targeting epidermal growth factor receptors in EGFR mutated non small lung cancer. Usually TKI are the first line treatment for advanced disease, and this issue has been examined deeply in several other reviews. However TKI much more recently have been tested in other clinical settings such as the adjuvant and neoadjuvant ones. These latter data are somehow contradictory.Author Response
Dear reviewer,
Thank you for you review and comments. We are honored to see that our manuscript filled your expectations, and we humbly hope this review will arouse the reader’s interest. You will find a revised version of the manuscript that we consider to be improved.
Regards.
Xavier Cansouline.
